# 2,6-diaminopurine promotes repair of DNA lesions under prebiotic conditions

Rafał Szabla [1,2 ✉], Magdalena Zdrowowicz [3 ✉], Paulina Spisz [3], Nicholas J. Green [4], Petr Stadlbauer [5], Holger Kruse [5], Jiří Šponer [5] & Janusz Rak [3]

High-yielding and selective prebiotic syntheses of RNA and DNA nucleotides involve UV irradiation to promote the key reaction steps and eradicate biologically irrelevant isomers. While these syntheses were likely enabled by UV-rich prebiotic environment, UV-induced formation of photodamages in polymeric nucleic acids, such as cyclobutane pyrimidine dimers (CPDs), remains the key unresolved issue for the origins of RNA and DNA on Earth. Here, we demonstrate that substitution of adenine with 2,6-diaminopurine enables repair of CPDs with yields reaching 92%. This substantial self-repairing activity originates from excellent electron donating properties of 2,6-diaminopurine in nucleic acid strands. We also show that the deoxyribonucleosides of 2,6-diaminopurine and adenine can be formed under the same prebiotic conditions. Considering that 2,6-diaminopurine was previously shown to increase the rate of nonenzymatic RNA replication, this nucleobase could have played critical roles in the formation of functional and photostable RNA/DNA oligomers in UV-rich prebiotic environments.

[1] EaStCHEM School of Chemistry, University of Edinburgh, Edinburgh, UK. [2] Institute of Physics, Polish Academy of Sciences, Warsaw, Poland. [3] Faculty of Chemistry, University of Gdańsk, Gdańsk, Poland. [4] MRC Laboratory of Molecular Biology, Cambridge, UK. [5] Institute of Biophysics of the Czech Academy of Sciences, Brno, Czech Republic. ✉email: rafal.szabla@ed.ac.uk; magdalena.zdrowowicz@ug.edu.pl

Exposure of nucleic acid strands to ultraviolet (UV) irradiation results in the formation of lesions which effectively suppress the key biological functions of DNA and may result in mutagenesis, immunosuppression, and skin cancer[1–3]. Cyclobutane pyrimidine dimers (CPDs) are among the most frequent photolesions and are formed in a [2 + 2] photocycloaddition reaction between the C=C bonds of adjacent pyrimidine bases[4–6]. While CPDs have a significant effect on the structure of the strand and its recognition, Nature developed complex machinery, i.e., photolyase enzymes, which locate the lesion and initiate the repair with photoinduced electron transfer from a flavin chromophore (see Fig. 1 and Supplementary Fig. 1 in the SI)[7,8]. However, photolyases evolved only after early living organisms emerged and, therefore, could not counteract photodamage formation in the very first oligonucleotides on Earth[9]. Therefore, more primitive molecular mechanisms of photodamage repair are likely to have protected the earliest biological systems from the ever-present threat of excessive UV-induced genetic mutation.

Recent advances in nonenzymatic template copying of nucleic acids[10–13] and prebiotic formation of their components[14–19] have brought us near to proposing a credible scenario for the emergence of informational polymers on early Earth. Among several different geochemical scenarios for the formation of purine and pyrimidine nucleosides[14–19], involvement of UV light offers highest selectivity by driving the key chemical transformations and destroying biologically irrelevant stereoisomers[20–24]. While high prebiotic UV fluxes could pass through the anoxic Archean atmosphere and support the synthesis and selection of key DNA and RNA building blocks[25], intense UV irradiation posed threat to the integrity of their oligomers. The resulting lack of compatibility between UV-assisted formation of nucleosides and the longevity of their oligomers remains one of the key unresolved issues for the origins of informational polymers on Earth. This led to the suggestion that specific DNA sequences could act as potent electron transmitters and allow photochemical charge separation leading to photoreversal of CPDs[26]. Nevertheless, the most photostable sequences like GATT, were demonstrated to repair only up to 25% of thymine CPDs (denoted as T=T)[26]. In addition, 8-oxo-purine bases, which could act as good electron donors and promote efficient photoinduced CPD repair[27], were shown to nearly completely inhibit nonenzymatic template copying of nucleic acids[13].

Previous theoretical works reported that charge-transfer (CT) electronic states in the GAT=T sequence are characterized by high vertical excitation energies and populated only indirectly, outside of the Franck–Condon region of the oligomer[28,29]. Therefore, involvement of an alternative nucleoside acting as a prebiotic flavin mimic could significantly lower the excitation energies of CT states and increase the self-repairing activity of oligonucleotides.

Here, we demonstrate that 2,6-diaminopurine (Dap) satisfies this criterion and enables highly efficient photoreversal of T=T CPDs. The subjects of our study were canonical and modified DNA trinucleotides containing a purine and two neighboring thymine (Thy) bases as model systems for photodamage repair. Consideration of trinucleotides consisting of deoxyribonucleosides is also consistent with the recent suggestion that DNA and RNA nucleosides could have emerged simultaneously on prebiotic Earth[24]. The self-repairing activity is manifested by higher T=T photoreversal quantum yields and electron transfer rates in the damaged trinucleotides containing Dap, when compared to canonical trimers containing adenine (Ade). It is worth noting that Dap is an excellent Watson–Crick base pairing partner for Thy and was shown to entirely replace Ade in the genome of phage S-2L of the cyanobacterium *S. elongatus*[30,31]. Furthermore, the deoxyriboside of Dap (D) was demonstrated to undergo prebiotic phosphorylation with diamidophosphate in the presence of 2-aminoimidazole, which could be followed by oligomerization with biological deoxyribosides[32]. Given that Dap could also enhance the rate of nonenzymatic RNA template copying[33–35], we provide a possible bridge between the UV-assisted synthesis of nucleotides and formation of their first self-replicating oligomers.

## Results

**Low energies of charge-transfer states lead to high efficiency of self-repair.** Calculations of vertical and adiabatic ionization energies (VIE and AIE) provide a good first estimate of electron-donating properties of nucleobases[36]. Our MP2 calculations indicated that the deoxyriboside of 2,6-diaminopurine (D) has lower VIE and AIE by over 1.8 eV when compared to canonical deoxyribosides of adenine (A) and guanine (G) (see Supplementary Table 3 in the SI). Consequently, we decided to computationally investigate the photophysical properties of DNA trinucleotides containing a CPD with two dimerized thymine bases (T=T) and D or A either on the 5′ or 3′ terminus. For this purpose, we carried out 20-μs long molecular dynamics (MD) simulations of these damaged trimers to identify their most representative stacked conformers and obtain their averaged structures (see Supplementary Figs. 21 and 22 in the SI). We further performed QM/MM calculations in an aqueous

**Fig. 1 Schematic representation of the self-repair mechanism of T=T dimers promoted by photoinduced electron transfer from 2,6-diaminopurine (D).** This mechanism is analogous to the enzymatic repair induced by the flavin chromophore acting as the electron donor (Supplementary Fig. 1 in the SI).

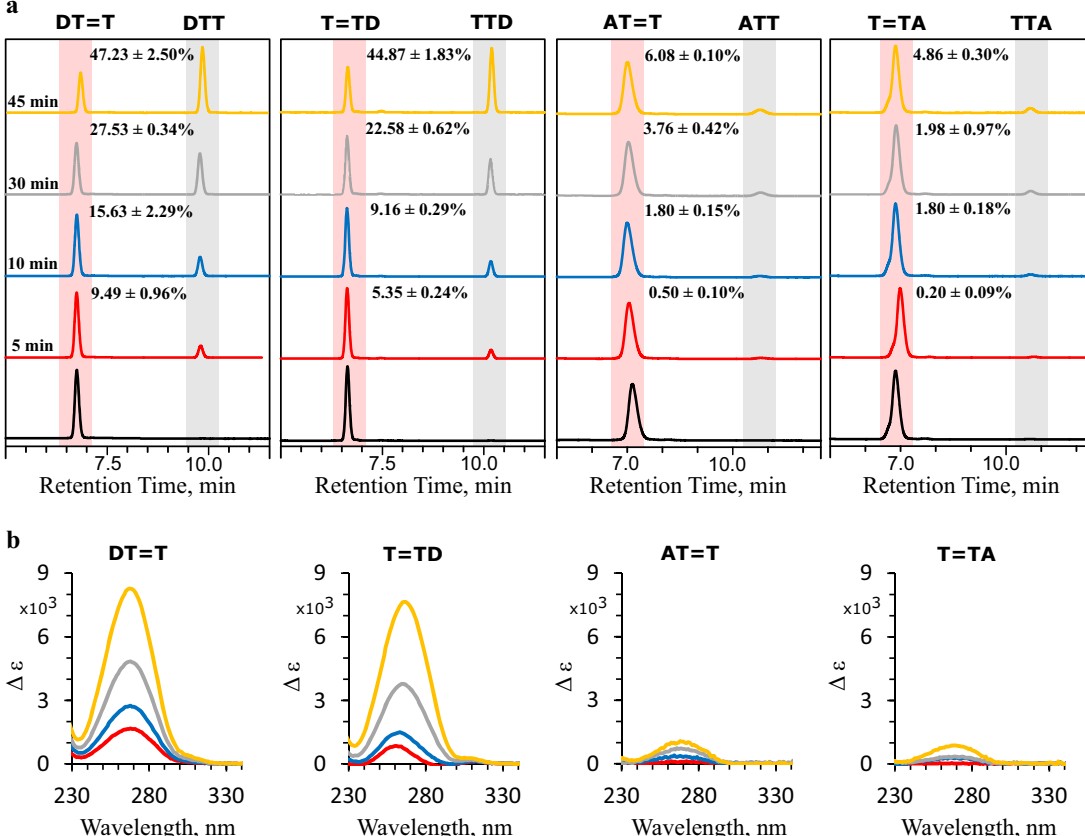

**Fig. 2 UV-irradiation experiments ($\lambda_{exc} = 280$ nm, $80\ \mu W\ cm^{-2}$) of damaged DNA trinucleotides containing a T=T dimer and A or D. a** Analytical HPLC analysis of irradiated DT=T, T=TD, AT=T, and T=TA performed after different irradiation times. The peaks at shorter retention times correspond to damaged trinucleotides, while the peaks at longer retentions correspond to the repaired photoproducts. **b** Differential absorption UV spectra demonstrating changes of the spectrum of the irradiated material at different irradiation times. Both trinucleotides containing D exhibit significant increase of absorption at 264 nm, which corresponds to the absorption maximum of repaired T chromophore. Only minor increase of absorption at 264 nm can be observed for the trinucleotides containing A.

environment with the algebraic diagrammatic construction to the second-order [ADC(2)][37,38] QM method, which was shown to provide accurate vertical excitation energies of the key photoactive states in DNA oligomers[39], and the TZVP basis set (see the "Methods" section for more details). As expected, the repairing $D^{\bullet+}T=T^{\bullet-}$ state has the excitation energy of 5.02 eV and exhibits configuration mixing with the bright (UV-absorbing) $\pi\pi^*$ state on D that is lower in energy by merely 0.14 eV. Even though the repairing $T=T^{\bullet-}D^{\bullet+}$ state has the vertical excitation energy of 5.62 eV, the bright $S_1(\pi\pi^*)$ state on D also has a notable contribution from this CT configuration, which implies good self-repairing potential. In contrast, the analogous CT states of AT=T and T=TA have significantly higher vertical excitation energies (5.74 and 5.80 eV, respectively) and do not exhibit any clear mixing with the low-energy $\pi\pi^*$ and $n\pi^*$ states on A (see Supplementary Table 4 in the SI).

Encouraged by these predictions we performed UV-irradiations of the four damaged trinucleotides at 280 nm and analyzed the mixture at different time intervals with HPLC experiments and UV-vis spectroscopy (Fig. 2a). At this wavelength, over 95% of absorbed photons excite the purine base, whereas the extinction coefficients of the CPDs are very small[40]. As shown in Fig. 2a, irradiations of trimers containing D lead to rapid formation of a photoproduct with corresponding HPLC peaks visible at longer retention times than the damaged trimers. The accumulation of this photoproduct at different irradiation times is proportional to the depletion of the starting

material. The differential absorption spectra of the UV-irradiated samples generated with respect to the starting materials (Fig. 2b) demonstrate a gradual increase of absorption at 264 nm, which corresponds to the absorption maximum of recovered Thy chromophores (see Fig. 1 for schematic representation of the self-repair process). In addition, we have validated by LC–MS that the photoproduct corresponds to the repaired Thy bases within the trimers (see section S2.1 in the SI).

After 45 min of continuous irradiation, DT=T and T=TD underwent thymine recovery in 47.2% and 44.9% yields, respectively (Fig. 2a). In contrast, equivalent exposure of AT=T and T=TA to UV light allowed for self-repair of only 6.1% and 4.9% of the corresponding starting materials. Therefore, presence of the additional amino group in the 2 position of the purine ring significantly enhanced the self-repairing properties. This is also reflected by an order of magnitude higher self-repair quantum yields observed for damaged trimers containing D (see Supplementary Table 2 in the SI). In reality, the overall photostability of specific DNA oligomers is the resultant of rates of various interplaying photoinduced phenomena, including formation of CPDs, self-repair, and direct benign photorelaxation channels of nucleobases.

To provide a better estimate of the photostability of our trinucleotides we performed continuous irradiations of their undamaged forms in the UV-B spectral range to reach the photostationary equilibria (Fig. 3). The irradiation of DTT reached the photostationary state in <20 min with only 8% of

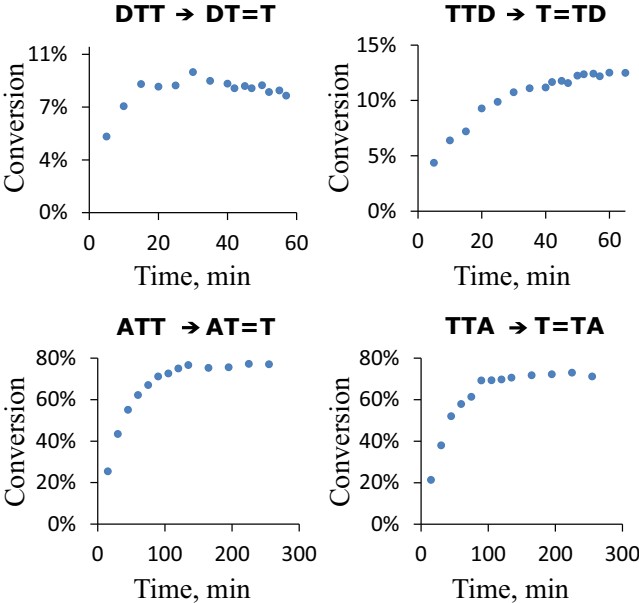

**Fig. 3 Continuous UV-irradiation experiments of undamaged DNA trinucleotides in the UV-B spectral range.** Samples were exposed to continuous UV-B flux (1.95 mW cm$^{-2}$) to yield photostationary equilibria of T=T dimer formation and self-repair. Analysis of photoproducts was performed by HPLC and % of conversion of undamaged DNA trimers into the damaged trinucleotides was calculated by integration of the respective peak areas (see the SI and the Methods section for more details).

CPDs formed, leaving over 90% of the trinucleotide intact (see Fig. 3 and Supplementary Fig. 18 in the SI). TTD also exhibits very high photostability with CPDs formed in <13% yield after the photostationary state is reached. On the contrary, continuous irradiations of ATT and TTA resulted in the formation of T=T dimers in 72% and 76% yields, respectively. This indicates that majority of the material containing the ATT and TTA sequences would be prevented from performing key prebiotically relevant functions such as self-replication.

**Molecular mechanism of self-repair in modified and canonical trimers.** To scrutinize the above differences in photoreactivity of the damaged trimers, we performed further QM/MM calculations with ADC(2) as the QM method and the TZVP basis set. As shown in Fig. 4a, direct electron transfer from UV-excited D to the T=T dimer is energetically favorable and the corresponding $S_1(CT)$ minimum lies ~1.5 eV lower in energy than the ring-puckered $S_1$ minimum of photoexcited D. The relative energies of the $D^{\bullet+}T=T^{\bullet-}$ and $D^*T=T$ excited-state minima with respect to the ground-state minimum (adiabatic excitation energies) amount to 2.95 and 4.42 eV, respectively. Apart from low vertical and adiabatic excitation energies of the repairing $D^{\bullet+}T=T^{\bullet-}$ state, electron transfer is facilitated by relatively high energy and sloped topography of the crossing between the $D^*T=T$ state and the ground state, which lies 0.4 eV above the respective $S_1$ minimum and hinders direct photorelaxation of D. Consequently, UV-excited D is much more likely to transfer an electron to the T=T dimer than undergo direct photorelaxation in DT=T. Similarly, the somewhat lower efficiency of self-repair observed for T=TD can be explained in terms of higher vertical excitation energy of the $T=T^{\bullet-}D^{\bullet+}$ state and less sloped topography of the $S_1/S_0$ state crossing[41] enabling smoother direct photorelaxation of D (Fig. 4b). The relatively high yields of self-repair in this trinucleotide are the consequence of a very low adiabatic excitation energy of the $T=T^{\bullet-}D^{\bullet+}$ state (2.52 eV). Finally, once the

negative charge is transferred to the T=T dimer, it induces breaking of the C5–C5 bond in the $S_1(CT)/S_0$ state crossing. The remaining C6–C6 bond of the CPD is opened spontaneously in the vibrationally hot electronic ground state[28], as schematically demonstrated in Fig. 1.

Negligible self-repairing activity of AT=T and T=TA trimers derives from high adiabatic and vertical excitation energies of the $A^{\bullet+}T=T^{\bullet-}$ and $T=T^{\bullet-}A^{\bullet+}$ states. In particular, the $S_1$ $T=T^{\bullet-}A^{\bullet+}$ minimum is lower in energy than the $S_1$ T=TA* minimum corresponding to photoexcited A base by merely 0.10 eV (see Fig. 4d). We were unable to locate the $S_1$ minimum for the $A^{\bullet+}T=T^{\bullet-}$ state, which is the result of an even higher adiabatic excitation energy for this state and lower energies of other valence excitations outside the Frank-Condon region of T=TA (see the SI for discussion). Furthermore, A may undergo much more efficient photorelaxation than D, which is manifested by a peaked $S_1/S_0$ state crossing separated from the respective $S_1$ minima by a modest energy barrier of merely 0.3 eV in both AT=T and T=TA trimers (Fig. 4c and d). It is worth noting that while our estimate is an upper bound to the barrier energy, this process is driven by ultrafast C2-H tilting motion, which can be facilitated by H atom tunneling.

As observed for canonical and undamaged DNA trimers and the GAT=T tetramer, the energies of various CT states are strongly affected by the structural arrangement of the oligomer[28,42]. In particular, Lee and Matsika[42] showed that excitation energies of CT states are directly proportional to the distance between the donor–acceptor pair. Similarly, we assign the very low vertical excitation energy of the CT state in the DT=T trimer to a practically perfect spatial alignment of the internal Thy and Dap bases and the short distance between them (see section S2.2 in the SI for detailed discussion). Consequently, higher vertical excitation energy of the CT state in the T=TD trimer (by ~0.6 eV) is consistent with longer distance between the internal Thy and Dap bases. However, we do not see a straightforward correlation between the distance separating the donor and acceptor bases and the adiabatic excitation energies of the $D^{\bullet+}T=T^{\bullet-}$ and $T=T^{\bullet-}D^{\bullet+}$ states (CT minima).

We provide further interpretation of our experimental observations based on calculations of approximate charge-transfer rates within the Marcus theory of electron transfer and considering couplings between locally excited states on purine bases and CT states, derived from the Boys localization approach[43] (see Supplementary Table 5 and the associated discussion in the SI for more details). The estimated rate constants calculated using the geometries and energies from Fig. 4 indicate that charge separation in photoexcited D-containing trimers occurs on a timescale of hundreds of femtoseconds and could indeed outcompete the direct photorelaxation of D (see Table 1). In contrast, the estimated timescale of electron transfer from excited A to T=T is over an order of magnitude longer than in the case of trimers containing D (~5.82 ps for T=TA). Considering that UV-excited A undergoes direct photodeactivation to the electronic ground state within <2.0 ps, its excited-state lifetime is too short to permit photoreversal of the T=T dimer[44].

**Self-repair of internal CPDs in longer nucleic acid oligomers.** The above results demonstrate that the photochemistry of damaged DNA trimers is characterized by the competition between purine photorelaxation and electron transfer processes and that the presence of D could induce efficient self-repair of CPDs on the termini of oligomers. To explore whether D could induce self-repair of internal CPDs, we performed analogous QM/MM simulations for the ADT=TA pentamer having an adenosine residue on each terminus. To examine the excited-state

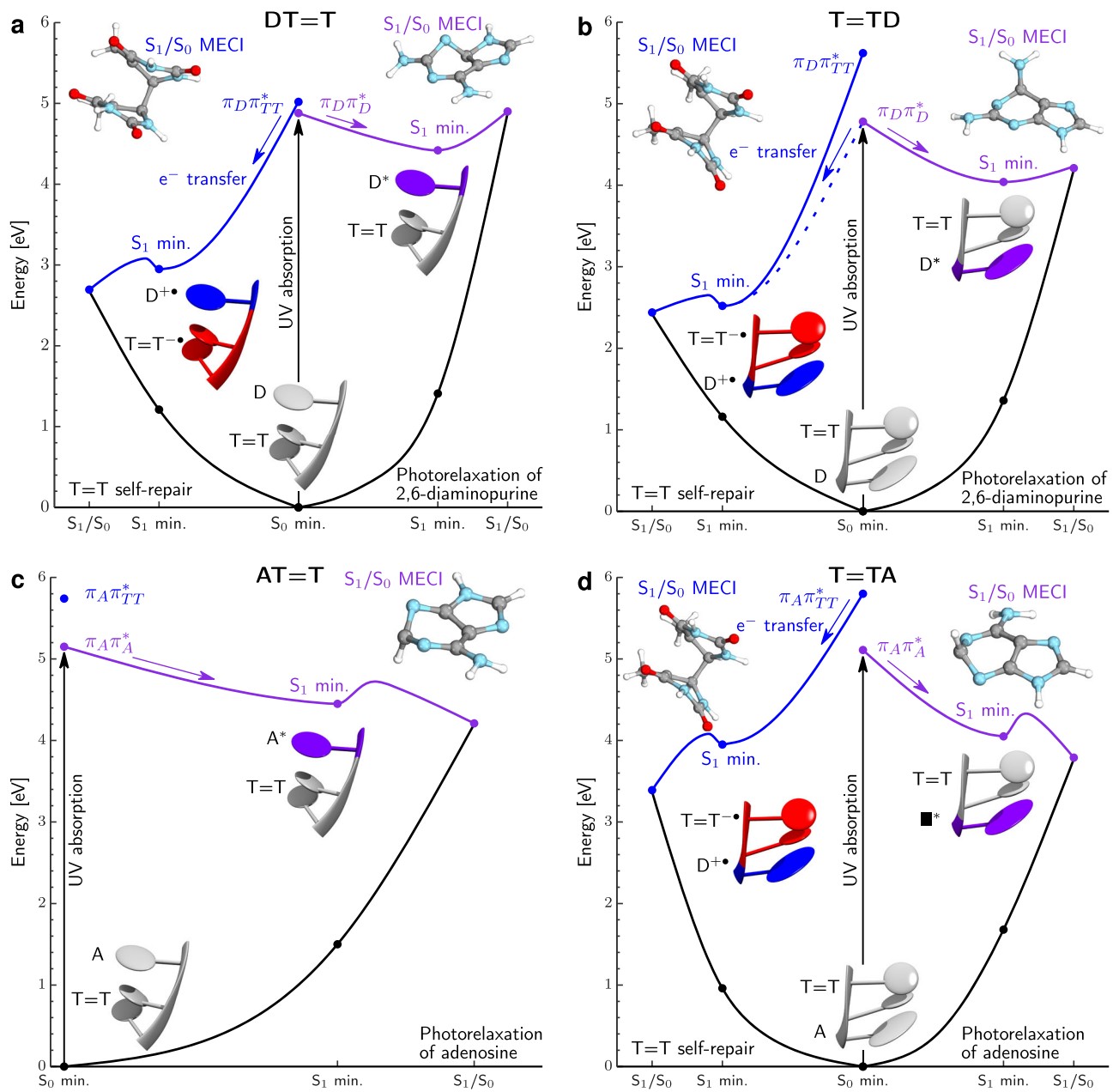

**Fig. 4 Potential energy profiles calculated with the QM$_{bases}$/MM approach for the key photorelaxation mechanisms of all four studied trinucleotides. a** Blue curve corresponds to self-repair of the T=T dimer triggered by the electron transfer from D. The C5–C5 bond is broken in the D$^{•+}$T=T$^{•-}$/S$_0$ state crossing. Direct photorelaxation of D base for DT=T – purple curve. Self-repair is the energetically more favorable process. **b** Analogous photorelaxation channels in T=TD. Dashed blue line indicates mixing between the T=T$^{•-}$D$^{•+}$ and T=TD$^*$ states. **c** The T=T$^{•-}$A$^{•+}$ configuration is not energetically available in the S$_1$ state and only the direct photorelaxation of A is presented. **d** High adiabatic excitation energy of the repairing A$^{•+}$T=T$^{•-}$ state results in dominant role of direct photorelaxation of A in AT=T. The electronic-structure (QM) calculations were performed at the ADC(2) and MP2 levels with the TZVP basis set. The molecular structures depict the key structural changes for purine bases or the T=T in the S$_1$/S$_0$ minimum-energy conical intersection (MECI).

**Table 1 Electron transfer rate constants and the corresponding timescales of electron transfer from the purine base to the T=T CPD estimated for three of the studied trinucleotides.**

| Trinucleotide | $k$ [s$^{-1}$] | $\tau$ [ps] |
|---|---|---|
| DT=T | $7.08 \times 10^{12}$ | 0.141 |
| T=TD | $3.85 \times 10^{12}$ | 0.260 |
| T=TA | $1.72 \times 10^{11}$ | 5.820 |

charge-transfer properties of ADT=TA we performed MD simulations and selected a conformer with all five nucleobases stacked, so that it mimics the structural arrangement of even longer single strands. We next performed calculations of vertical excitation energies and optimizations of the key minima of the S$_1$ excited state at the ADC(2)/def2-SVP level (note that a smaller basis set was applied because of increased computational costs). The lowest-lying excited state calculated for the optimized ground-state geometry of ADT=TA has the energy of 4.90 eV and can be characterized as the bright (UV-absorbing) $\pi\pi^*$ excitation on D mixed with the AD$^{•+}$T=T$^{•-}$A electronic

configuration responsible for self-repair (see Supplementary Fig. 26 and the associated discussion in the SI for more details). In addition, the $S_1$(CT) minimum of the $AD^{\bullet+}T{=}T^{\bullet-}A$ state is lower in energy than the $S_1$ minimum of the locally excited state on D by 0.72 eV. This value, referred to as the driving force of electron transfer, is approximately two times lower in magnitude than in the case of the DT=T and T=TD trimers. However, the CT minimum is still considerably lower in energy than the $AD^*T{=}TA$ ($\pi\pi^*$) minimum, which together with the low vertical excitation energy of the $AD^{\bullet+}T{=}T^{\bullet-}A$ ($\pi_D\pi_{TT}^*$) excitation shows that electron transfer from D to the T=T is also an energetically favorable process in the studied pentamer. Therefore, we expect the stacked conformer of ADT=TA to be involved in highly efficient self-repair of its internal CPD.

Our QM/MM simulations indicate that excited-state electron transfer between the D and T=T fragments is comparably as efficient inside a well-stacked DNA single strand as in the investigated DNA trimers. However, according to the classical MD simulations and conformational analysis, the ADT=TA oligomer spent <5% of time in the fully stacked arrangement and it sampled various partly or fully unstacked conformations for the remaining part of the simulation. In contrast, the stacked conformer of AT=T was populated for more than 50% of time during analogous simulations (see the SI for more details). Therefore, any potential irradiation experiments on the ADT=TA damaged pentamer would largely reflect the averaged photoreactivity of random coil structural arrangements. While such random coils would be present on the early Earth, the results would be less conclusive toward internal CPDs. In contrast, longer and more stable DNA double strands exhibit very complex photochemistry with multiple possible photoproducts[26], owing to a large number of stacked chromophores. Consequently, we focused here only on short single-stranded oligomers, to clearly disentangle CPD self-repair from other competing photorelaxation channels. Self-repair in a flexible single-stranded DNA oligomer could also operate in partly unstacked conformations. We expect that conformers having a sufficiently close distance between the Dap base and the T=T dimer, and unstacked flanking bases, exhibit very similar photochemistry to the studied DNA trimers. Further experimental work is underway to test these predictions.

**Prebiotically plausible synthesis of β-2,6-diaminopurine 2′-deoxyriboside.** Considering its photostabilizing activity in DNA trimers, we sought to determine whether β-2,6-diaminopurine 2′-deoxyriboside (β-D) might be available on a prebiotic Earth for incorporation into nucleic acids. Recently, Xu et al.[22] reported a proof of principle synthesis of deoxyribosides, including deoxyadenosine, by transglycosylation of 2′-deoxy-2-thiouridine (2tU) with adenine, to provide 4% and 6% of β- and α-A, respectively. The synthesis, although low yielding, demonstrates that a prebiotic world that contained RNA may also have contained small amounts of deoxyadenosine. Since Dap has previously been synthesized in a prebiotic fashion[45], and has even been discovered on meteorites[46], we reasoned that it may also have accumulated and participated in this transglycosylation reaction. Indeed, after evaporating an aqueous mixture of 2tU and Dap (15 equiv.) and heating to 100 °C for 40 h, β-D was formed (2%) along with at least one isomer (probably the α-diastereomer, 4%) and other minor compounds (see Fig. 5). Thus, the reaction proceeds similarly to that of the reported synthesis of deoxyadenosine, providing similar proof of principle that DNA containing Dap may have indeed been present in a prebiotic world.

**Photostability of D vs A.** UV irradiation was suggested as the key environmental factor, which selected adenosine, guanosine,

**Fig. 5 Transglycosylation reaction of 2′-deoxy-2-thiouridine (2tU) furnishing 2,6-diaminopurine 2′-deoxyriboside (D).** Transglycosylation of 2tU in the dry state with D gives a mixture of α and β stereoisomers of the D deoxyriboside alongside loss of the 2tUra base.

cytidine, uridine, and thymidine as the most photostable canonical nucleosides[47]. Therefore, we performed comparative continuous irradiations of D and A within the UV-C spectral domain, to establish whether D could survive in the prebiotic environment as a potential building block. Our HPLC analysis (Supplementary Fig. 19 in the SI) performed for different irradiation times demonstrates that D undergoes photodegradation approximately two times more rapidly than A when exposed to UV-C radiation (254 nm). Approximately 11.6% and 5.9% of D and A were decomposed, respectively, after 120 min of continuous irradiation (Supplementary Fig. 19 in the SI). Even though D undergoes photodegradation more rapidly than A, the difference in photostability in the UV-C range is not overwhelming. Analogous irradiation experiments performed in the UV-B spectral range resulted in the photodegradation of D in merely 16.4% after 120 min, whereas A remained practically photostable under these conditions and its depletion was below the accuracy threshold of our analyses (Supplementary Fig. 20 in the SI). It worth noting that the measured photon flux in the UV-B region was an order of magnitude higher than in the UV-C range and several times higher than the overall UV flux estimated for the early Earth (see the following section for more details). Therefore, we conclude that D is sufficiently photostable to have been involved in the formation of the first informational polymer.

Based on our ADC(2) calculations, we ascribe the lower photostability of D to the fact that its key photorelaxation mechanisms are associated with out-of-plane rotation of amino groups that require longer timescales than the ultrafast C2-H bond rotation in A[48]. Thus, despite a modest energy barrier, rotation of the C2-H bond occurs more rapidly in aqueous environment than rotation of the C2-NH$_2$ group, which forms hydrogen bonds with nearby water molecules. Furthermore, the C2-NH$_2$ rotation in D leads to a sloped $S_1/S_0$ state crossing that lies 0.3 eV above the C2-puckered $S_1$ minimum (see Fig. 6). This additionally hinders damageless photodeactivation of D. Consequently, the anticipated longer lifetimes of the electronically excited bi-radical state in D are likely to entail destructive bimolecular photoreactions.

**Comparison of irradiation conditions and prebiotic UV fluxes.** The integrated surface flux between 200 and 300 nm delivered to the surface of early Earth was estimated to be ~2700 erg s$^{-1}$ cm$^{-2}$ [23]. Therefore, we aimed to irradiate our samples with UV fluxes that would be consistent with this value within an order of magnitude in terms of intensity. Our tunable UV lamp used for the irradiation of damaged DNA trimers at 280 nm delivered flux of 800 erg s$^{-1}$ cm$^{-2}$, which is in good agreement with the UV environment expected for prebiotic Earth. In this experiment, we focused on irradiating our samples with the same wavelength applied previously by Pan et al.[40]

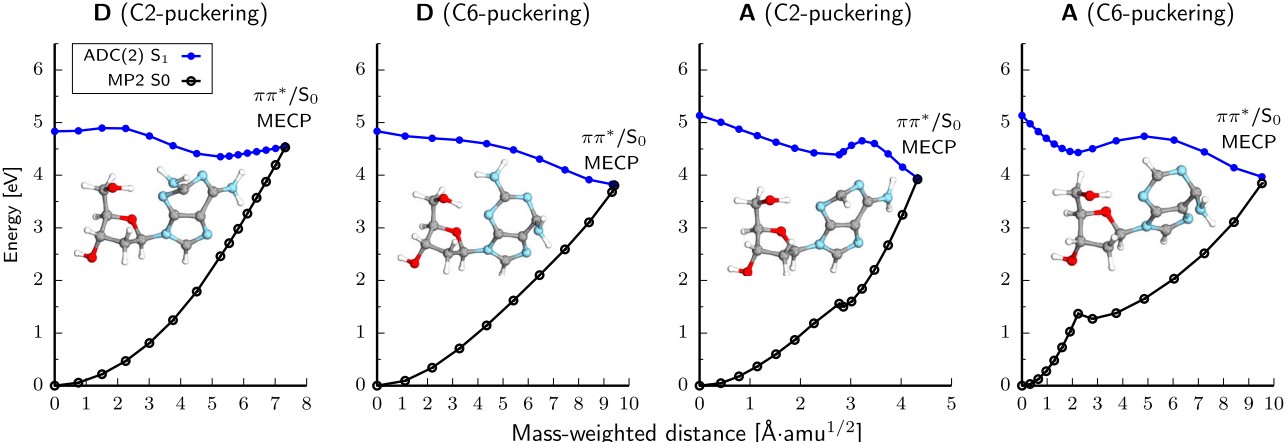

**Fig. 6 Potential energy profiles for the key photorelaxation mechanisms of 2,6-diaminopurine and adenine 2′-deoxyribosides (D and A).** The electronic-structure calculations for ground and excited states were performed at the MP2 and ADC(2) levels, respectively, using the TZVP basis set. MECP minimum-energy crossing point.

in analogous irradiation experiments of canonical DNA trimers. Irradiation experiments aiming to determine the photostationary equilibria of undamaged DNA trimers and the photostability of D and A nucleosides delivered higher fluxes of 17,000 erg s$^{-1}$ cm$^{-2}$ in the UV-B spectral range. It is important to emphasize that even though we did not perform broadband irradiation experiments that would fully cover the spectrum of the young Sun, our work provides a good comparative assessment of the self-repair properties and photostability of trinucleotides and nucleosides comprising Ade and Dap bases. Consequently, data presented in Fig. 3 and Supplementary Fig. 19 indicate that the D nucleosides and the studied trimers containing D could withstand days of continuous exposure to UV light on early Earth. Broadband irradiation experiments aiming to estimate the half-life of short D-containing DNA oligomers in UV-rich prebiotic environments are currently underway.

## Discussion
In summary, we have demonstrated that D might have served as an important component of the primordial genetic alphabet by significantly enhancing the photostability of DNA oligomers and increasing the self-repair rates of cyclobutane pyrimidine dimers (CPDs). These effects result from a decreased energy threshold for photoinduced charge separation and enhanced efficient intrastrand electron transfer. In particular, D readily inserts an electron into a CPD which is followed by self-repair of the damaged site. Thus, D can be considered as a prebiotic flavin mimic enabling DNA self-repair with ~90% yield. To further reinforce its prebiotic relevance, we have shown that D could have been formed on early Earth alongside A through a transglycosylation reaction. Alternative syntheses of purine (deoxy)nucleosides raise the prospect that D might also be a product of these alternative chemistries, perhaps in more significant yields[15,17,24,49,50]. Given its only moderately lower photostability relative to A, D could survive long enough in a UV-rich environment to be incorporated into oligomers and play a further role in enzyme-free DNA self-replication. Ongoing questions remain as to the reason and mechanism by which D was eventually removed from the genetic alphabet. It is worth noting that the additional amino group in the 2 position of the purine ring could affect functional tertiary structures of RNA or hinder separation of duplexes. Consequently, if D had provided limited functionality compared to, for example, A, it might have not persisted after the emergence of the first CPD-repairing enzyme. The overarching and unintuitive result of this study is that the formation of highly photostable molecular assemblies could be achieved by joining moderately photostable or even photounstable building blocks, here represented by D and pyrimidine-rich DNA fragments. We anticipate that these findings will shed further light on UV-induced prebiotic selection, as it has been widely postulated that Nature selected only the most photostable building blocks to protect their biomolecular polymers from the photodamage caused by continuous and intense UV irradiation[47]. The result of this work could be also used to support the design of self-healing macromolecules[51] or to enhance the electron transfer properties of DNA[52] through the design of photostable synthetic XNAs.

## Methods
### Irradiation experiments
*Samples.* All oligonucleotides (ATT, TTA, DTT, TTD) were purchased from Future Synthesis, Poland. The samples were HPLC-purified by the Future Synthesis. 2′-deoxyadenosine and 2,6-diaminopurine 2′-deoxyriboside were purchased from Merck and Alfa Aesar, respectively. All solutions were prepared using ultra-pure water produced by a Millipore purification system.

### Generation of AT=T, T=TA, DT=T, T=TD
*Illumination conditions.* The thymidine photodimerization was conducted in the pH 7.0 buffer solution of 10 mM phosphate containing 100 mM sodium chloride. All samples were illuminated with UV-B light in quartz tubes with the use of Irradiation Chamber BS-04 (Opsytec Dr. Gröbel, Ettilngen) at an average power of 1.95 mW measured by a Maestro11 power/energy meter, equipped with an 11PD photo-detector (Standa, Lithuania). The accumulation of photoproduct was performed using HPLC (see Supplementary Fig. 4) followed by characterization with the use of mass spectrometry (LC–MS and LC–MS/MS, see Supplementary Figs. 5–16).

*HPLC analysis.* Damaged trinucleotides were isolated using reversed-phase HPLC method (Dionex Ultimate 3000 System with a Diode Array Detector, which was set at 260 nm). The mobile phases used in this study were 0.1% HCOOH and 80% ACN in water (gradient elution from 0 to 25% ACN in 15 min) with a flow rate of 1 mL/min. The separations were conducted on the Wakopak Handy ODS column (4.6 × 150 mm, 5 μm in particle size, and 100 Å in pore size).

*Mass spectrometry.* Ultra-High Performance Liquid Chromatography system, Nexera X2 (Shimadzu) was coupled to ESI-TripleTOF 5600+ (AB SCIEX) mass spectrometer operated in negative ion mode. The separations were performed under the following conditions: Kinetex column (Phenomenex C18, 100 Å, 2.1 × 150 mm); flow rate 0.3 mL/min; a gradient elution with 80% ACN and 0.1% HCOOH in water as mobile phases (from 0 to 25% ACN in 15 min). The MS experiments included the following parameters: −4.5 kV ion spray voltage (ISV); −80 V declustering potential (DP); −10 V collision energy (CE); 200 °C source temperature.

### Self-repair of T=T lesions
*Illumination conditions.* All samples were illuminated at 280 nm in quartz capillaries with the use of a xenon lamp. The 280-nm wavelength of incident light was selected using a monochromator (M250 Optel, Poland). The average incident light intensity for the above irradiation setup was 80 μW. This value was determined with a Maestro11 laser power/energy meter, equipped with an 11PD photodetector

(Standa, Lithuania). To provide stable measuring conditions at pH 7, the oligo-nucleotides were dissolved in water and buffered by phosphate buffer at a con-centration of 10 mM. For each illumination experiment, the concentration of trinucleotides was equal to 20 μM. Illumination was performed in different time intervals up to 45 min. Between the illumination steps absorbance spectra from 230 to 340 nm were recorded. All experiments were performed at least in triplicate and followed by HPLC analysis and characterization of repair product by mass spec-trometry (LC–MS and LC–MS/MS).

*HPLC analysis.* Irradiated and non-irradiated samples of trinucleotides were ana-lyzed using the method described in the section "Generation of AT=T, T=TA, DT=T, T=TD".

*Mass spectrometry.* All analyses were performed using the method described in the section "Generation of AT=T, T=TA, DT=T, T=TD".

## Photodegradation of D and A

*Illumination conditions.* The water solution of A and D ($10^{-4}$ M) containing phosphate buffer (10 mM, pH = 7.0) was illuminated at UV-C radiation in quartz capillaries for 30, 60, 90, and 120 min. All samples were deoxygenated before irradiation by purging with argon and illuminated in Irradiation Chamber BS-04 (Opsytec Dr. Gröbel, Ettilngen) used at an average power of 170 μW (UV-C) or 1.7 mW (UV-B) measured by a Maestro11 power/energy meter, equipped with an 11PD photodetector (Standa, Lithuania). All experiments were performed at least in duplicate and followed by HPLC analysis. The percentage of photodegradation was calculated by integration of the respective peak areas.

*HPLC analysis.* Irradiated and non-irradiated samples of A and D were analyzed with the use of reversed-phase HPLC method on Dionex UltiMate 3000 System with Diode Array Detector. The detection of the effluents was carried out at 260 nm. Separations were performed using a Wakopak Handy ODS column (C18, 4.6 × 150 mm, 5 μm; 100 Å), 0.1% HCOOH and 80% ACN in water as mobile phases (gradient elution from 0 to 10% ACN in 10 min) and a flow rate of 1 mL/min.

## Synthetic procedures

*Solid-state glycosylation of 2,6-diamonopurine (Dap) with 2′-deoxy-2-thiouridine (2tU) to form β-2,6-diaminopurine 2′-deoxyriboside (β-D).* 0.20 mL of a standard solution (22 mM in MeOH) of 2′-deoxy-2-thiouridine 2tU (1.0 mg, 4.4 μmol, 1.0 equiv.) was evaporated in a 10 mL round-bottom flask, leaving 2tU as a white solid distributed on the flask. 2,6-Diaminopurine Dap (9.3 mg, 66 μmol, 15.0 equiv.), and milliQ-filtered water (pH 8 by addition of traces of NaOH) (0.9 mL) were added and the suspension was sonicated until all the 2′-deoxy-2-thiouridine had been dissolved (the diaminopurine remains insoluble). The pH at this stage was mea-sured at 8. This mixture was then concentrated on a rotary evaporator at 45 °C and ~5 mbar until the solvent had been evaporated, leaving a white deposit on the inside of the flask. The flask was then immersed in an oil bath at 100 °C. After 40 h, the material was dissolved in D₂O and 1H NMR spectra recorded. The supernatant of this mixture was used for HPLC and LC–MS analysis, showing the presence of mainly diaminopurine Dap, 2-thiouracil 2tUra, and small quantities of β-2,6-dia-minopurine 2′-deoxyriboside (β-D) and an isomer, presumably the α-configured diastereomer (a standard of this material is not available, however, the reaction appears to have a similar outcome, and similar retention times, to the reported reaction between 2′-deoxy-2-thiouridine 2tU and adenine which forms α- and β-configured diastereomers. The isomer has the same molecular mass and an almost identical UV-Vis spectrum to β-D). The identity of β-2,6-diaminopurine 2′-deoxyriboside β-D was confirmed by spiking experiments and measuring by both NMR and HPLC (Figs. S2, S3). The yield could not be determined accurately by NMR because of overlapping signals. Development of an analytical HPLC method (see below) gave conversions after 40 h of 2% and 4% for β-D and the presumed α-diastereomer, respectively. Analytical HPLC was used to monitor the reaction, using a Thermofisher Ultimate 3000 UPLC and Waters Atlantis T3 C18 column (5 μm, 4.6 mm × 150 mm). UV absorption calibration curves for standards of 2′-deoxy-2-thiouridine 2tU, thiouracil 2tUra, β-2,6-diaminopurine 2′-deoxyriboside β-D were collected. UV absorption for known peaks (verified by spiking) was recorded and calibration curves used to determine relative amounts. Recovery was calibrated to aliquots from time zero at which the yield of starting material 2′-deoxy-2-thiouridine 2tU was set to 100%.

## Molecular dynamics simulations.
Force-field parameters for the T=T dimer were taken from our previous study[28]. Both the AT=T and T=TA trinucleotides were solvated in a truncated octahedral water box with the distance between solute and box border being at least 10 Å. For each trinucleotide we used two different water models, SPC/E[53] and OPC[54]. The box was first neutralized by 2 K⁺ ions and then 0.15 M KCl was added, using Joung and Cheatham parameters for SPC/E and TIP4Pew, respectively[55]. The simulations were carried out with the parmOL15 force field[56], which contains several dihedral reparametrizations[57–59] of the DNA force field developed by Cornell et al.[60]. Standard equilibration protocol was employed first. Five hundred steps of steepest descent minimization were followed by 500 steps of conjugate gradient minimization with 25 kcal mol⁻¹ Å⁻² position restraints on

DNA atoms. We then imposed position restraints of 25 kcal mol⁻¹ Å⁻² on the DNA fragment and heated the system from 0 to 300 K over 100 ps. Heating was performed at constant volume. The system was then minimized with 5 kcal mol⁻¹ Å⁻² restraints on the DNA oligonucleotide using 500 steps of steepest descent method followed by 500 steps of the conjugate gradient optimization approach. The whole system was then equilibrated for 50 ps at a constant pressure of 1 bar and constant temperature of 300 K. We further performed an analogous series of consecutive minimizations and equilibrations applying decreasing restraints on the atomic position of the DNA fragment of 4, 3, 2, and 1 kcal mol⁻¹ Å⁻². The final equili-bration was carried out using position restraints of 0.5 kcal mol⁻¹ Å⁻² and starting velocities from the previous equilibration step and was followed by free (unrest-rained) molecular dynamics simulation for 50 ps. Temperature and pressure cou-pling constants were set to 0.2 ps during the equilibration, and to 5 ps during the final phase of molecular dynamics. Production phase was started afterward and each system was simulated for 10 μs. The time step was set to 4 fs with application of the hydrogen mass repartitioning[61] and the SHAKE[62] and SETTLE[63] algorithms. The temperature and pressure were set at 300 K and 1 bar, respectively, using Berendsen weak-coupling thermostat and barostat[64]. Electrostatic interactions were treated using the Particle Mesh Ewald method (PME). The non-bonded cutoff was set to 9 Å. The calculations were performed with the CUDA accelerated pmemd module of AMBER 16[65].

Each trajectory was clustered to find the important conformers. We applied a modified15 algorithm of Rodriguez and Laio[66], and used eRMSD as a metric[67]. It is a nucleic acid-specific measure that determines the similarity of two structures based on relative orientation of its nucleobases and is a suitable tool for distinguishing stacked and unstacked conformations or syn- and anti-oriented nucleotides. Only clusters with a population exceeding 1% were selected for further inspection.

## QM/MM simulations

*Preparation of starting solvated oligonucleotide structures.* After analyzing and clustering the trajectories simulated for the AT=T and T=TA trimers we selected the most representative stacked conformers for the QM/MM simulations (see Supplementary Figs. 21 and 22 and the accompanying discussion). These con-formers have the structural features of longer oligonucleotides and therefore our qualitative observations and results of the excited-state calculations are also rele-vant for longer DNA fragments. The stacked structural arrangement of AT=T is also the highest populated conformer in both MD simulations using the SPC/E and OPC solvent models. Even though the most populated conformer of T=TA is an unstacked structure, the stacked conformer is among the most populated structures of this trinucleotide (see Supplementary Figs. 21 and 22). In order to perform QM/MM calculations on equivalent structures of native trimers and those modified with 2,6-diaminopurine (Dap) we substituted the C2-H position of adenine with an amino group and generated the corresponding DT=T and T=TD stacked struc-tures. Each trinucleotide was then solvated in a sphere of explicit water molecules with the radius of 24 Å. The solvent was then equilibrated for 10 ps with position constraints imposed on the solute molecule (trinucleotide). The resulting geome-tries were used in subsequent ground-state geometry optimizations.

*QM/MM setups.* The QM/MM calculations were performed in two different par-titioning setups involving either the whole trinucleotide in the QM region (QM$_{DNA}$/MM setup) or just the nucleobases treated at the QM level (QM$_{bases}$/MM setup), as presented in Fig. 7. The remaining part of the system was described with the same force-field suite as in the MD simulations, including the SPC/E[53] force field for water. We applied the link hydrogen atom scheme in the QM$_{bases}$/MM setup with the MM charge of the link H atom being set to 0. The boundary between the QM$_{bases}$ and MM regions bisected the N-glycosidic bonds. We employed the electrostatic embedding framework to all of the QM/MM calculations. We per-formed these simulations using our in-house modification of the QM/MM interface[68] of the AMBER suite of programs[65] to enable QM calculations employing the TURBOMOLE 7.3 program[69].

*Geometry optimizations.* We optimized the minimum-energy geometries of the four selected structures of damaged DNA trimers using the cost-effective com-posite PBEh-3c[70] hybrid DFT scheme for the electronic-structure calculations and the QM$_{DNA}$/MM setup. The PBEh-3c method was recently demonstrated to pro-vide accurate geometries and energies of short nucleic acid fragments[71]. Initial optimizations of the ground-state geometries were performed for the whole sol-vated system using the AMBER limited-memory BFGS algorithm. We performed tighter optimization of the internal region of the sphere (within the radius of 12.5 Å from the most central atom of each conformer) using the open-source external geometry optimizer XOPT, which employs the rational function and approximate normal coordinates schemes[72,73]. The outer region of the sphere was kept frozen during this optimization procedure. This approach was used for all ground-state and excited-state geometry optimizations.

*Excited-state QM/MM calculations.* Calculations of excited-state energies and gradients within the QM/MM framework were performed using the algebraic diagrammatic construction to the second-order [ADC(2)][37,38,74] method for electronic-structure calculations, whereas corresponding energies and gradients for

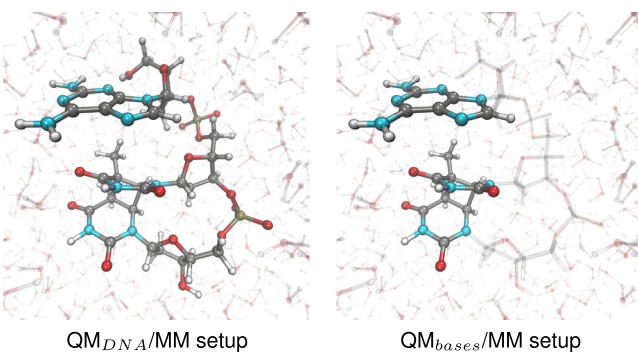

QM$_{DNA}$/MM setup
DFT-D3 optimizations;
ADC(2)/TZ energies

QM$_{bases}$/MM setup
(only nucleobases in QM)
ADC(2)/TZ optimizations

**Fig. 7 QM/MM setups used in the geometry optimizations and energy calculations.** Solid balls and sticks models show the two considered QM regions, while transparent residues (water molecules in both setups and the sugar-phosphate backbone on the right) correspond to the MM region described at the force-field level. The MM region interacts with the QM part via point charges (electrostatic embedding). ADC(2)/TZ denotes the ADC(2) method with the TZVP basis set.

the electronic ground-state were obtained at MP2 level. All of the ADC(2)/MP2 calculations were performed using the TZVP basis set. Vertical excitation energies contained in the Supplementary Table 4 were calculated using the QM$_{DNA}$/MM setup on top of the geometries optimized at the PBEh-3c level. Optimizations of different minima on the S$_1$ potential energy surface and S$_1$/S$_0$ minimum-energy crossing points (MECPs) were performed using the QM$_{bases}$/MM setup. The orbital characters of excited electronic states were determined based on analysis of molecular orbitals and key contributing configurations. In addition, we established the charge-transfer numbers and performed electron-hole population analysis using the TheoDORE 1.5.1 package[75,76]. MECP optimizations were performed using the algorithm of Levine, Coe, and Martínez[77], which allows to locate highly representative geometries of S$_1$($\pi\pi^*$)/S$_0$ conical intersections. Even though the single-reference ADC(2)/MP2 approach does not provide a correct description of the entire topography of S$_1$/S$_0$ conical intersections, it was demonstrated that it can accurately describe geometries of S$_1$($\pi\pi^*$)/S$_0$ MECPs in biomolecular systems with conjugated $\pi$ bonds including qualitatively correct shapes of PE surfaces from the Frank-Condon region up to the state crossing[78,79]. While much less accurate results were obtained for S$_1$($\pi\pi^*$)/S$_0$ conical intersections[79], in this work, we only considered S$_1$($\pi\pi^*$)/S$_0$ MECPs with only one dominant electronic configuration contributing to the excited-state character. Therefore, we expect that our approach provides at least qualitatively correct description of the mechanism. The energies of the key stationary points (S$_0$ and S$_1$ minima, MECPs, and vertical excitation energies) were used to construct illustrative PE surfaces in Fig. 3. The barrierless profiles between the Franck–Condon region of each trinucleotide and different S$_1$ minima were confirmed by excited-state geometry optimizations. The barrier heights between the charge-transfer S$_1$ minima and corresponding MECPs responsible for C5–C5 bond cleavage were estimated based on a relaxed potential energy scan along the C5–C5 distance performed for the whole trimer (see Supplementary Fig. 25 for more details). The barrier heights (presented in Fig. 3) between the S$_1$ minima of the purine bases and the corresponding S$_1$/S$_0$ MECPs were taken from the calculations performed for single nucleosides.

**Simulations of the ADT=TA pentamer.** Classical molecular dynamics simulations and subsequent ground-state geometry optimizations within the QM/MM framework for the ADT=TA pentamer were performed in an analogous way to the AT=T and T=TA trimers. The excited-state QM/MM calculations for this pentamer were also performed in an analogous way to the studied DNA trimers (using the ADC(2) method), except that we used the smaller def2-SVP basis set. See also the final section of the Supplementary Information and Supplementary Fig. 26, for more details.

**Excited-state QM simulations of A and D nucleosides.** The excited-state energies and geometries for adenosine and 2,6-diaminopurine nucleosides were calculated in the gas phase using the ADC(2) method for electronic excitations. The ground-state geometries and energies were obtained at the MP2 level. Similarly as in the case of the QM/MM calculations, for trinucleotides we used the TZVP basis set to keep the methodology consistent. MECPs for isolated nucleosides were optimized using the same algorithm as described above. The potential energy profiles presented in Fig. 5 in the main article were constructed by means of linear interpolation in internal coordinates between the respective S$_0$ geometries (Frank-Condon region), S$_1$ geometries, and S$_1$/S$_0$ MECPs. All of the gas phase electronic-structure calculations were performed with the TURBOMOLE 7.3 program[69].

## Data availability
The authors declare that the data supporting the findings of this study are available within the paper and its Supplementary Information files.

## Code availability
The external optimizer XOPT is available under the link https://github.com/hokru/xopt. The modified interface between the TURBOMOLE 7.3 and AMBER 16 programs, which enabled performing QM/MM simulations with the ADC(2) electronic-structure method was deposited on the Figshare platform under the link https://figshare.com/s/b8a5526e0c3335a270ab with https://doi.org/10.6084/m9.figshare.14268056. Most recent revision of this code can be obtained upon reasonable request.

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

## Acknowledgements

The authors thank Prof. John D. Sutherland, Prof. Dimitar D. Sasselov, and Dr. Corinna Kufner for helpful discussions. This research was supported by the Simons Foundation (494188 to R.S.), the Foundation for Polish Science (START Fellowship to R.S.), the Czech Science Foundation (project 21-23718S to J.S. and P.St.), the project SYMBIT (reg. number: CZ.02.1.01/0.0/0.0/15_003/0000477 to J.S. and H.K.) financed by the ERDF and a grant from the Ministry of Science and Higher Education in Poland (DS/531-T080-D494-20 to J.R.).

## Author contributions

M.Z., P.S., and J.R. irradiation experiments; N.J.G. nucleoside synthesis; P.St. and J.S. molecular dynamics simulations; H.K. code development; R.S. idea, quantum chemical simulations, and coordination of the project. R.S. wrote the main article with input from all coauthors. M.Z., N.J.G., P.St., and R.S. assembled the SI and "Methods" section.

## Competing interests

The authors declare no competing interests.
