## [Peer Review File · Nature Communications]

Reviewers' Comments:

Reviewer #1:

Remarks to the Author:

The manuscript reports that diaminopurine containing DNA trinucleotides DT=T and T=TD lead to high self repair yields of the T=T cyclobutane dimers (reaching 92%) relative to the trinucleotides that contain A in place of D. This is a very topical finding given recent progress in the areas of prebiotic nucleotide monomer syntheses and their nonenzymatic copying and replication. As the latter areas, which appeared generally intractable in the not too distant past, accumulate plausibly prebiotic solutions, focus has turned to additional critical questions such as how photodamaged DNA and RNA might be repaired nonenzymatically. 8-Oxo-G is a nonstandard nucleobase that was explored several years ago as a possible flavin analogue. Despite its initial promise in promoting repair of thymidine photodimers, it has subsequently been shown not to function in nonenzymatic copying of nucleic acids, making it an implausible prebiotic constituent. In contrast, diaminopurine riboside has a long established ability to function well in nonenzymatic template-directed reactions in place of adenosine (beyond even the two papers cited in the manuscript). Although not cited, it is perhaps notable that diaminopurine fully replaces adenine in phage S-2L of the cyanobacterium *S. elongatus*. The self healing properties of diaminopurine modified trinucleotides constitute a critical new finding for UV-mediated origins scenarios. However, the generality and importance of diaminopurine facilitated self healing needs further assessment before it could be referred to as a missing link without qualifications (per the abstract). In the context of the present manuscript the authors should examine at least two examples where there is a neighboring base on both sides of diaminopurine as this constitutes the minimal model for repair of an internal cyclobutane pyrimidine dimer. Without such a neighbor, the present system strictly models repair of CPD's on the termini of oligomers, and is suggestive of what may occur internally.

Reviewer #2:

Remarks to the Author:

This paper presents a study on the ability of 2,6-diaminopurine (Dap) to repair the photochemical product CPD in DNA. Overall, the study is very strong. It offers experimental proof that Dap repairs CPD upon irradiation. It also offers a theoretical explanation for why that is happening. So, I think the work provides important information to the questions about prebiotic formation of DNA. A broader discussion on the uniqueness of Dap should be included in order to justify publication to Nature.

There are other bases, such as guanine and 8-oxoguanine, that the authors mention can also repair CPD. What about other unnatural bases? What is unique about Dap compared to other bases that could also be formed in prebiotic conditions? Why did the authors choose this molecule? Do they think other bases can also satisfy this role?

Minor comments:

The authors should give some rationale for why the charge transfer states behave differently in the DTT vs TTD and ATT vs TTA. In particular the difference in DTT vs TTD is quite large (5eV vs 5.6 eV). This difference in 5' vs 3' neighboring bases has been explained before for longer oligonucleotides (Faraday Discussions, 216, 507, (2019)) based on differences in the distances between T and purine bases. Is the argument similar here? Why are the CT state energies different? And how would that be affected by the flexibility of the trinucleotides?

There should be some discussion in SI about the inadequacy of ADC(2) to describe conical intersections with the ground state.

Reviewer #3:

Remarks to the Author:

In their manuscript, Szabla et al provide strong evidence that the non-canonical nucleobase 2,6-

diaminopurine (Dap) may have contributed to increasing the UV-resistance of DNA-polymers on early Earth. Using single-stranded trinucleotides containing TT dimers as model system, the authors show that the electron donating properties of Dap allow the direct photorepair of cyclobutane pyrimidine dimers (CPDs), the dominant form of photolesions caused by UV-irradiation. Compared to adenine, Dap causes higher quantum yields and electron transfer rates in the damaged trinucleotides, which enables CPD photoreversal. Since Dap is compatible with non-enzymatic copying of genetic information, it might have protected early biological systems from lethal photodamage before the emergence of photorepair enzymes. Finally, the authors provide also experimental evidence for a potential prebiotic pathway of β -2,6-diaminopurine 2'-deoxyriboside formation from 2'-deoxy-2-thiouridine and Dap.

This study combines thorough in silico calculations / predictions with very solid experimental results from UV-vis spectroscopy, HPLC and MS. It offers a credible solution to the rather paradox situation that UV is required for nucleoside/nucleotide formation on the one hand, but is harmful to all other downstream steps of primitive information transfer on the other.

Main comments:

- An illustration of the general CPD repair pathway in the introduction would be very helpful for readers that are not familiar with DNA photorepair (I guess there are quite a few in the field). For example by moving Figure S1e to the main text (possibly replacing "FADH" with a more general term).
- Figure 1a: Are there time points beyond 45 minutes for this data? Are the photostationary states from Fig S19 also reached in these experiments / does the data converge?
- The repair yields of 92% mentioned in the abstract are somewhat hidden in Figure S19. I think this data belongs into the main text. Plotting ATT vs DTT and / or TTA vs TTD in a single graph each could look quite convincing.
- The yields of the β -2,6-diaminopurine 2' deoxyriboside synthesis based on the transglycosylation protocol of Xu et al. from 2019 are very low. Do the authors expect that also a tethered glycosylation of α -anhydropyrimidines by 2,6-diamino-8-mercaptadenine would also work (similar to ref 24)? If so, it might be worth mentioning this e.g. in the conclusion section.
- It would be advantageous to move the comparison between the irradiation conditions used in this study and the UV environment on early Earth from the SI to the main text. Also, mentioning the $\mu\text{W}/\text{cm}^2$ somewhere in the main text or in the figure legends would be very helpful.
- If Dap was so crucial to the survival of primitive genetic material on early Earth, why was it later replaced by A? A brief speculation as to why this might have happened would reinforce the proclaimed prebiotic relevance of the paper.
- Why was the photodamage of trimers investigated with UV-B but the photostability of isolated A/DAP monomers performed with UV-C?
- How would a DNA-duplex geometry influence Dap's ability to repair CPDs? I would assume that there are fewer conformers possible and additional electronic coupling between stacked bases. Stable stacking might also lead to the formation of delocalized exciton states upon UV absorption. Can the authors comment on this? Of course, any potential computational or experimental data would add greatly to the paper.
- Taking the decreased photostability of D into account: how many days would a DTT/TTD trimer block survive under credible early Earth conditions?

Minor comments:

- please add page numbers to the SI.
- please check references again. E.g. ref 13 at line 200 is certainly not correct.
- early Earth not earth.
- It would be helpful to explain in the figure legend why the percentages in Fig. 1a do not correlate

with the relative peak areas of the substrates / products.

Reviewer #1 (Remarks to the Author):

*The manuscript reports that diaminopurine containing DNA trinucleotides DT=T and T=TD lead to high self repair yields of the T=T cyclobutane dimers (reaching 92%) relative to the trinucleotides that contain A in place of D. This is a very topical finding given recent progress in the areas of prebiotic nucleotide monomer syntheses and their nonenzymatic copying and replication. As the latter areas, which appeared generally intractable in the not too distant past, accumulate plausibly prebiotic solutions, focus has turned to additional critical questions such as how photodamaged DNA and RNA might be repaired nonenzymatically. 8-Oxo-G is a nonstandard nucleobase that was explored several years ago as a possible flavin analogue. Despite its initial promise in promoting repair of thymidine photodimers, it has subsequently been shown not to function in nonenzymatic copying of nucleic acids, making it an implausible prebiotic constituent. In contrast, diaminopurine riboside has a long established ability to function well in nonenzymatic template-directed reactions in place of adenosine (beyond even the two papers cited in the manuscript). Although not cited, it is perhaps notable that diaminopurine fully replaces adenine in phage S-2L of the cyanobacterium *S. elongatus*. The self healing properties of diaminopurine modified trinucleotides constitute a critical new finding for UV-mediated origins scenarios. However, the generality and importance of diaminopurine facilitated self healing needs further assessment before it could be referred to as a missing link without qualifications (per the abstract). In the context of the present manuscript the authors should examine at least two examples where there is a neighboring base on both sides of diaminopurine as this constitutes the minimal model for repair of an internal cyclobutane pyrimidine dimer. Without such a neighbor, the present system strictly models repair of CPD's on the termini of oligomers, and is suggestive of what may occur internally.*

Response: We thank Reviewer #1 for their supportive comments and assessment of our manuscript. We have added further citations describing the possible role of 2,6-diaminopurine base (Dap) in prebiotic chemistry as indicated above (see new references: 29-31,48 and 49). We also addressed the question of Dap-facilitated CPD repair longer oligomers or larger molecular contexts. For this purpose we performed analogous molecular dynamics simulations of the ADT=TA pentamer, which represents a minimal system containing internal CPD lesion and D nucleotide (and not at the termini). These simulations and subsequent clustering of structures with respect to root mean squared displacement RMSD, returned over 20 conformers for both water models (SPC/E and OPC). Most importantly, most conformers corresponded to largely unstacked and disordered structures with low populations. Similarly, the populations of stacked conformers did not exceed 5%. It is worth noting that many of these conformers had a close contact between the D and T=T moieties with unstacked terminal purines, which would enable electron transfer and self-healing to occur in a similar manner to the DT=T and T=TD trimers. These results indicate that irradiation experiments performed for similar pentamers containing internal CPDs would not provide definitive answer about CPD self-repair in larger molecular contexts. While this might be possible for various double-stranded DNA fragments, such systems are also characterized by a much broader range of competing photochemical and photophysical processes. These processes are often difficult to distinguish with simple irradiation experiments and HPLC separation. Finally, there is a very broad range of possible sequences that could be considered here, with various nucleotides neighbouring the D and T=T fragments. Therefore, we decided to limit our investigations of Dap-facilitated CPD repair in this work to studying how the two neighbouring Ade bases could affect the electron transfer properties of the ADT=TA pentamer by means of excited-state QM/MM calculations. For this purpose, we selected the fully stacked conformer of this pentamer that best resembles strand conformation in double-stranded B-DNA helix.

Our calculations indicate that similarly to the DT=T and T=TD trimers, the ADT=TA pentamer may also undergo efficient electron transfer from the Dap base to the CPD lesion. We added an additional sub-section to the Results and Discussion entitled 'Self-repair of internal CPDs in longer nucleic acid oligomers' with two paragraphs which describe these computational results and their implications.

Finally, as indicated by Reviewer #1 we also modified the last sentence of the abstract and introduction in order to better reflect the findings presented in our work.

Reviewer #2 (Remarks to the Author):

This paper presents a study on the ability of 2,6-diaminopurine (Dap) to repair the photochemical product CPD in DNA. Overall, the study is very strong. It offers experimental proof that Dap repairs CPD upon irradiation. It also offers a theoretical explanation for why that is happening. So, I think the work provides important information to the questions about prebiotic formation of DNA. A broader discussion on the uniqueness of Dap should be included in order to justify publication to Nature.

There are other bases, such as guanine and 8-oxoguanine, that the authors mention can also repair CPD. What about other unnatural bases? What is unique about Dap compared to other bases that could also be formed in prebiotic conditions? Why did the authors chose this molecule? Do they think other bases can also satisfy this role?

Response: We thank Reviewer #2 for their supportive comments. Indeed, we mentioned in the introduction that 8-oxoguanine could act as efficient electron donor. We further clarified this fragment in order to better describe the challenges in promoting self-repair with 8-oxo-G and canonical nucleobases. The fragment from the second paragraph of the introduction now reads:

“Nevertheless, the most photostable sequences like GATT, were demonstrated to repair only up to 25% of thymine CPDs (denoted as T=T)²⁶. In addition, 8-oxo-purine bases, which could act as good electron donors and promote efficient photoinduced CPD repair²⁷, were shown to nearly completely inhibit nonenzymatic template copying of nucleic acids¹³.”

This fragment also clarifies that most efficient CPD self-repair in canonical oligonucleotides was demonstrated for the GAT=T tetramer (Ref. 26). The same work alongside Ref. 36 also showed that single canonical purines (such as guanine) are very inefficient in transferring electron to the CPD lesion upon photoexcitation. To underline the uniqueness of Dap, we have also modified the introduction to show that it could replace Ade in specific genomes by selective pairing with T. The last part of the introduction emphasizes that Dap could enhance nonenzymatic RNA template copying (see lines 79-87). We also cited a recent prebiotic phosphorylation and oligomerization study (Ref. 31), which considered the D nucleoside. Finally, in the Conclusions section, we suggested a possible scenario showing how Dap could have been eventually replaced by A as suggested by Reviewer #3.

Minor comments:

The authors should give some rational for why the charge transfer states behave differently in the DTT vs TTD and ATT vs TTA. In particular the difference in DTT vs TTD is quite large (5eV vs 5.6 eV). This difference in 5' vs 3' neighboring bases has been explained before for longer oligonucleotides (Faraday Discussions, 216, 507, (2019)) based on differences in the distances between T and purine bases. Is the argument similar here? Why are the CT state energies different? And how would that be affected by the flexibility of the trinucleotides?

Response: We thank for this very important comment and for pointing out the mentioned article. Indeed the distance and alignment of the purine base and CPD moiety are the key factors, which determine the energies of CT states and consequently, electron transfer rates. In particular, in the case of the system promoting most efficient self-repair (DT=T trinucleotide) we noticed very good

alignment between the purine ring of the Dap base and the C4=O carbonyl group of the neighbouring T base involved in the CPD, which hosts the molecular orbital accepting the transferred electron. This alignment is much less effective in the T=TD trimer and the distance between the Dap base and T=T dimer is larger in this system, leading to higher energies of CT states. We have added another brief paragraph summarising these observations to the section ‘Molecular mechanism of self-repair in modified and canonical trimers’ and referred to similar observations for canonical undamaged trinucleotides described in the article suggested by the Reviewer (lines 191-201 and newly added ref. 41). We also added a more detailed description of the distances between the donor and acceptor bases to the SI and, referred to existing figures and to the vertical and adiabatic excitation energies of the CT states.

There should be some discussion in SI about the inadequacy of ADC(2) to describe conical intersections with the ground state.

Response: We thank the Reviewer for pointing this out to us. We added the following fragment to the Methods section in the SI and cited additional references to support our claim:

“Even though the single-reference ADC(2)/MP2 approach does not provide a correct description of the entire topography of S_1/S_0 conical intersections, it was demonstrated that it can accurately describe geometries of $S_1(\pi\pi^*)/S_0$ MECPs in biomolecular systems with conjugated π bonds including qualitatively correct shapes of PE surfaces from the Frank-Condon region up to the state crossing^{30,31}. While much less accurate results were obtained for $S_1(n\pi^*)/S_0$ conical intersections, in this work, we only considered $S_1(\pi\pi^*)/S_0$ MECPs with only one dominant electronic configuration contributing to the excited-state character. Therefore, we expect that our approach provides at least qualitatively correct description of the mechanism.”

Reviewer #3 (Remarks to the Author):

In their manuscript, Szabla et al provide strong evidence that the non-canonical nucleobase 2,6-diaminopurine (Dap) may have contributed to increasing the UV-resistance of DNA-polymers on early Earth. Using single-stranded trinucleotides containing TT dimers as model system, the authors show that the electron donating properties of Dap allow the direct photorepair of cyclobutane pyrimidine dimers (CPDs), the dominant form of photolesions caused by UV-irradiation. Compared to adenine, Dap causes higher quantum yields and electron transfer rates in the damaged trinucleotides, which enables CPD photoreversal. Since Dap is compatible with non-enzymatic copying of genetic information, it might have protected early biological systems from lethal photodamage before the emergence of photorepair enzymes. Finally, the authors provide also experimental evidence for a potential prebiotic pathway of β -2,6-diaminopurine 2'-deoxyriboside formation from 2'-deoxy-2-thiouridine and Dap.

This study combines thorough in silico calculations / predictions with very solid experimental results from UV-vis spectroscopy, HPLC and MS. It offers a credible solution to the rather paradox situation that UV is required for nucleoside/nucleotide formation on the one hand, but is harmful to all other downstream steps of primitive information transfer on the other.

Main comments:

- *An illustration of the general CPD repair pathway in the introduction would be very helpful for readers that are not familiar with DNA photorepair (I guess there are quite a few in the field). For example by moving Figure S1e to the main text (possibly replacing “FADH” with a more general term).*

Response: We thank Reviewer #3 for their supportive comments and remarks. To address the first main comment we shifted Fig. 2 to the introduction and modified the caption. It now is denoted as Fig. 1. This way, the figure appears much earlier in the text and allows the readers to immediately see the schematic representation of CPD repair. While this figure demonstrates the self-repair mechanism induced by the Dap base, it is analogous to the DNA repair mechanism triggered by the photoexcited flavin chromophore. Therefore, to avoid showing the same information twice and to save space for other figures, we decided to leave Figure S1 in the previous form in the SI. However, we now refer to Figure S1 also in the caption to the new Figure 1.

• *Figure 1a: Are there time points beyond 45 minutes for this data? Are the photostationary states from Fig S19 also reached in these experiments / does the data converge?*

Response: Indeed, 45 min (Figure 2 in the revised manuscript) was the longest irradiation time, which is too short to produce a photostationary state as indicated by Figure S19. Due to low efficiency of generating the damaged trimers (by irradiation of the native ones) we limited the analysis of repair to only four time points (see Figure 2). In our experiments, each time point requires a newly prepared solution. Thus, the more irradiation times are studied, the more material is needed. The main purpose behind Figure 2 is to show much higher yields of self-repair for trimers comprising the D nucleotide, alongside differential UV absorption spectra showing the increase of UV absorption in the region expected for repaired thymine bases. On the other hand, the photo-stationary states were determined for the reverse processes e.g. ATT \rightarrow AT=T, where the amount of material for irradiation was limited only by the cost of the order from the oligonucleotide synthesis company. Let us note, that the photodamage and self-repair processes are the dominant photochemical processes in the studied trinucleotides, consequently the states of photostationary equilibrium can be reached both from the native as well as photodamaged structures.

• *The repair yields of 92% mentioned in the abstract are somewhat hidden in Figure S19. I think this data belongs into the main text. Plotting ATT vs DTT and / or TTA vs TTD in a single graph each could look quite convincing.*

Response: According to the Referee suggestion, we plotted TTA/TTD data in a single graph (see Figure R1 below). Since the photostationary states are reached at different times and conversion rates such plots are less aesthetically appealing and the data points for D-containing oligomers are more congested than for the canonical trimers (Figure R1). In our opinion, the presentation of stationary state data for all systems, as in Figure S19, is clearer. Therefore, the data shown in Figure S19 has been moved (as the Referee suggests) to the main text as a new figure (Fig. 3).

Figure R1. Estimation of photostationary states of photodimerization in the studied systems.

• *The yields of the β -2,6-diaminopurine 2' deoxyriboside synthesis based on the transglycosylation protocol of Xu et al. from 2019 are very low. Do the authors expect that also a tethered glycosylation of α -anhydropyrimidines by 2,6-diamino-8-mercaptoadenine would also work (similar to ref 24)? If so, it might be worth mentioning this e.g. in the conclusion section.*

Response: We thank R3 for this insight, and have, as per the suggestion, included a sentence in the conclusion that highlights our synthesis of D may be improved by this new DNA nucleoside synthesis, or indeed related purine ribonucleoside syntheses:

“Alternative syntheses of purine (deoxy)nucleosides raise the prospect that D might also be a product of these alternative chemistries, perhaps in more significant yields^{15,17,24,48,49}.”

• *It would be advantageous to move the comparison between the irradiation conditions used in this study and the UV environment on early Earth from the SI to the main text. Also, mentioning the $\mu\text{W}/\text{cm}^2$ somewhere in the main text or in the figure legends would be very helpful.*

Response: We added information about UV fluxes for specific experiments to the figure captions. We also added a brief sub-section about the comparison between our irradiation conditions and the UV environment on the early Earth before the conclusions section (see lines 325-341).

• *If Dap was so crucial to the survival of primitive genetic material on early Earth, why was it later replaced by A? A brief speculation as to why this might have happened would reinforce the proclaimed prebiotic relevance of the paper.*

Response: We thank for this comment. Indeed, we have been considering the same question when working on this article. While the answer to this question is not obvious and still a speculation, we indicated that the additional amino-group in the 2 position of the purine ring would affect functional tertiary structures or hinder strand separation. More importantly, even though our work offers a solution to the problem of emergence of nucleic acid oligomers in UV-rich prebiotic environments, the prebiotic relevance of the D nucleoside will have to be validated by more research, which could demonstrate the transition to a four-letter genetic alphabet. We added three sentences commenting on this issue in the Conclusions section (see lines 355-359 in the revised manuscript).

• *Why was the photodamage of trimers investigated with UV-B but the photostability of isolated A/DAP monomers performed with UV-C?*

Response: We agree with Reviewer #3 that we used different wavelength ranges for the photodamage and photostability. The main reason for this was that both purine nucleosides underwent very slow photodegradation within the UV-B spectral range. To address this issue we performed additional irradiation experiments and added Figure S21, which depicts photostability of D and A studied in the UVB range. These results concern only the D nucleoside because A was photostable under UVB irradiations (the obtained differences in the peak areas for irradiated and nonirradiated A samples were within the error of HPLC method). The appropriate paragraph describing experimental conditions has been added to the Materials and methods section. We also modified

• *How would a DNA-duplex geometry influence Dap's ability to repair CPDs? I would assume that there are fewer conformers possible and additional electronic coupling between stacked bases. Stable stacking might also lead to the formation of delocalized exciton states upon UV absorption. Can the authors comment on this? Of course, any potential computational or experimental data would add greatly to the paper.*

Response: We agree with Reviewer 3 that this is a very important question. To address this, we added a full paragraph describing the energetics of electron transfer process in a longer ADT=TA oligomer with all bases stacked. Since this model represents a single DNA strand, we took particular care to perform the excited-state QM/MM calculations on a representative and stacked conformer that would also resemble strand conformation in B-DNA. Consequently, this gives us further confidence that good electron donating properties of Dap could also enable CPD self-repair in larger molecular contexts. We included an additional section in both the main article and the SI, which discuss the simulations performed for this pentamer. As indicated in the answers to Reviewer #2, electronic couplings and energies of CT states are strongly dependent on the structural arrangement and distance between the donor and acceptor sites. Therefore, we anticipate that DNA double helix could facilitate efficient electron transfer from D owing to its less flexible stacked conformation, which maintains this distance short. However, there is a large variety of structural motifs and sequences, which should be explored in the future, to provide a definitive answer to this question.

• *Taking the decreased photostability of D into account: how many days would a DTT/TTD trimer block survive under credible early Earth conditions?*

Response: We thank Reviewer #3 for raising this very interesting question. We commented on this matter in the final paragraph before the Conclusions section (lines 325-341). In brief, it is very difficult to infer the half-lives of DTT/TTD trimers based on our irradiation experiments, performed in selected UV spectral ranges. This is caused by the fact that the rates of competing photochemical processes are wavelength dependent. Here, we focused on irradiations of the DNA trimers at 280 nm to reproduce previous measurements of photoinduced self-repair efficiency in canonical trimers performed by Pan et al. (*J. Phys. Chem. B* **116**, 698–704 (2012), ref. 36). UV-irradiation in the UV-B spectral range allowed us to more easily reach the photostationary equilibria, but also covered only a limited range of wavelengths that could be delivered to the surface of early Earth. The main goal of this work was to perform comparative irradiation experiments of canonical vs. modified DNA building blocks to show the enhancement of photostability and self-repair upon the incorporation of the D nucleoside. However, as noticed by the Reviewer, our experiments indicate that the half-lives of DTT and TTD in UV-rich prebiotic environments would reach days. Future broadband irradiations with an Xe arc lamp, which spectrum resembles the spectrum of the young Sun, could provide a much more accurate estimate.

Minor comments:

- *please add page numbers to the SI.*
- *please check references again. E.g. ref 13 at line 200 is certainly not correct.*
- *early Earth not earth.*
- *It would be helpful to explain in the figure legend why the percentages in Fig. 1a do not correlate with the relative peak areas of the substrates / products.*

Response: We thank for noticing these issues. We applied all of the corrections suggested above.

Reviewers' Comments:

Reviewer #1:

Remarks to the Author:

I believe the authors have adequately addressed my reviewer comments in their revised manuscript.

Reviewer #2:

Remarks to the Author:

I am satisfied with the authors response to the comments.

Reviewer #3:

Remarks to the Author:

The authors have addressed my comments satisfactorily and I support the publication of this article.